# Equus β-Defensin-1 Regulates Innate IMMUNE Response in *S. aureus*-Infected Mouse Monocyte Macrophage

**DOI:** 10.3390/ani12212958

**Published:** 2022-10-27

**Authors:** Le Pei, Kun Liu, Wei Wei, Hong Su, Feng Li, Ying Feng, Daqing Wang, Xiunan Li, Yongyue Hou, Guifang Cao

**Affiliations:** 1Inner Mongolia Key Laboratory of Basic Veterinary Medicine, College of Veterinary, Inner Mongolia Agricultural University, Huhhot 010010, China; 2Inner Mongolia Academy of Agricultural and Animal Husbandry Sciences, Huhhot 010010, China; 3School of Public Healthy, Inner Mongolia Medical University, Huhhot 010010, China

**Keywords:** β-defensin-1, Equus, *S. aureus*-infected, immune response

## Abstract

**Simple Summary:**

The β-defensin-1 (BD-1) is rich in disulfide bonds and antibacterial peptides with direct bactericidal function activity. Equine BD-1 was initially reported in 2004, however, there are no reports on the specific mechanisms that characterize the impact of Equine BD-1 on innate immune function. In this study, we map the tissue distribution of Equus BD-1 (i.e., Equine BD-1, *ass BD-1*, and *mule BD-1*) and compare their expression levels in various tissues, and we showed that Equine BD-1, *ass BD-1*, and *mule BD-1* have an identical (100%) open reading frame (ORF). Next, we expressed the ORF of Equus BD-1 sequence fragment (OEBD-1) in the *E. coli* expression system and used this to treat an *S. aureus*-infected murine macrophage cell line in vitro. Results indicated that the OEBD-1 increased cytokines expression and associated signaling pathway, furthermore, OEBD-1 promoted macrophage phagocytose *S. aureus* in vitro. The report provides a theoretical basis for the development of defensins as potential alternatives to antibiotics.

**Abstract:**

Beta-defensin-1 (BD-1) is among the class of antibacterial peptides that are rich in disulfide bonds, have direct antibacterial activity and showed enhanced expression following external stimulation. However, existing research studies only treated BD-1 to cell models without stimulation from pathogen-associated molecular patterns (PAMPs), which will further influence our understanding of the role of BD-1. In this study, we map the tissue distribution of Equus BD-1 (i.e., *eBD-1*, *ass BD-1*, and *mule BD-1*) and compare their expression levels in various tissues. We further characterize the three kinds of Equus BD-1 by analyzing their full-length cDNA. We showed that *eBD-1*, *ass BD-1*, and *mule BD-1* have an identical (100%) open reading frame (ORF). The ORF encoding OEBD-1 expressed the ORF in the *E. coli* Top10 expression system. This expression system was combined with an *S. aureus*-infected J774A.1 macrophage cell line to determine the influence on innate immune mediator expression. Using this expression model system, it was determined that the OEBD-1 protein enhanced IL-6 and TNF-α secretion. It can also promote *TLR2*, *IL-1β*, *CCL2*, *CCL7*, *CXCL10* and *NF-κB p65* mRNA expression. Moreover, OEBD-1 upregulates phosphorylation of ATK, Syk and IκB-α. In addition, OEBD-1 enhances the macrophage’s ability to phagocytose *S. aureus*. In conclusion, Equus BD-1 was shown to play an essential role in macrophage-involved innate immune responses in an in vitro system.

## 1. Introduction

Defensins are common antimicrobial cationic peptides containing 29–42 amino acids, including 6–8 cysteines [1,2]. β-defensin-1 (BD-1) is a family of antimicrobial peptides with 38–42 amino acids produced by a range of epithelial cells [3,4,5]. BD-1 is widely distributed and expressed in mammalian tissues. Previous work demonstrated that the human BD-1 (HBD-1) is abundantly expressed in the nasal mucosa, lungs, and gland epithelium [6,7,8]. Furthermore, the equine β-defensin-1 (*eBD-1*) is expressed mainly in the heart, spleen, liver, kidney, and intestine [9]. Davis et al. also showed that the cDNA sequences of *eBD-1* are 64.4% identical to human β-defensin-2 (HBD-2), suggesting that β-defensin is conserved across mammals [9].

β-defensins’ (BDs) expression reveals that BDs may link the innate and adaptive immune systems. The research showed that murine BD-2 evokes various cytokines in dendritic cells [10]. In addition, in some cell models, HBD-2 and human β-defensin-3 (HBD-3) have been shown to regulate inflammatory factors IL-6, IL-37, TNF-α, and IL-1 [11,12,13,14]. In contrast to the expression of α-, and θ-defensins, Selsted and Ouellette showed that BDs expression is inducible at the transcriptional level in many tissues [15], suggesting that BDs play a regulatory role after PAMPs’ induction, in host cells. However, most of references only treated BDs without stimulating cells in vitro, which might influence the further understanding of BDs. 

Equine BD-1 (*eBD-1*) is widely expressed and distributed in a range of equine tissues. Recent reports have shown that *eBD-1* is mildly chemotactic for equine neutrophil and monocytes during equine herpesvirus infection [16]. However, whether *eBD-1* affects macrophage function has not been reported. Macrophages are important immune cells for innate immune responses, which eliminate pathogenic microorganism-infected hosts by the secretion of cytokines and phagocytosis of PAMPs [17]. Cytokine expression and phagocytosis involves various signal transduction pathway within macrophages. The spleen tyrosine kinase (Syk) and PI3K-AKT are essential constituent elements for macrophage phagocytosis to occur [18,19,20]. 

In this study, we map the tissue distribution of Equus BD-1 (i.e., *eBD-1*, *ass BD-1*, and *mule BD-1*) and compare their expression levels in various tissues. We further characterize the three Equus BD-1 by analyzing their full-length cDNA. To determine whether the Equus BD-1 is involved in the innate immune response, we expressed the OEBD-1 in an *E. coli* expression system binding with elastin-like protein V_30_ (ELP-V_30_). We tested the OEBD-1 recombinant protein whether it could enhance macrophage phagocytic ability in *Staphylococcus aureus*-infected J774A.1 macrophage cell line and secretion of inflammatory cytokines, and explored the associated signaling pathway. Here, we demonstrated how OEBD-1 enhanced innate immune responses in *S. aureus*-infected macrophages in vitro.

## 2. Materials and Methods

Three adult male Mongolia horses (aged 18–20 months), three adult Dezhou male donkeys (aged 19–22 months), and three adult male mules (Mongolia X Dezhou donkeys, aged 19–24 months) were obtained from a local abattoir. None of the experiment animals had clinical signs of parasitic or infectious disease. The method of animal euthanasia was following Jin et al. [21], whereby the animals were given an overdose of the proprietary euthanasia solution Euthasol (pentobarbital sodium 100 mg/kg and phenytoin sodium 10 mg/kg), and the fresh mucosa tissue of the tongue, esophagus, stomach, and intestines, as well as tissues of the liver, lung, kidney, and spleen were harvested. This study was approved by the Institutional Animal Care and Use Committee of the Inner Mongolia Agricultural University (License No. SYXK, Inner Mongolia, 2014-0008) with adherence to Inner Mongolia Agricultural University guidelines.

### 2.1. cDNA Preparation

Tissue RNA extractions were carried out using the Axygen RNA kit (Axygen Scientific, Union City, CA, USA), following the manufacturer’s instructions. First, forty mg of cultured tissues were placed in 400 µL lysis buffer (Axygen Scientific, USA) and flash-frozen in liquid nitrogen. RNAs were extracted from tissue by ferrite bead crushing.

Cellular RNA extractions were performed following the manufacturer’s instructions. Cells were resuspended by 100 uL lysis buffer, and centrifugalized at 12,000× *g* with subsequent RNA extraction as per the manufacturers’ instructions. The RNA (A260/A280 value between 1.9–2.0) at a 100 ng/μL concentration and 5 μL volume was reverse-transcribed into cDNA using the Primer Script TM RT Master Mix Kit (Takara, Dalian, China). 

### 2.2. PCR

PCR primers were designed using the equine, ass, and mule nucleotide information derived from the National Center for Biotechnology Information (https://www.ncbi.nlm.nih.gov, accessed on 1 June 2022) (Table 1). The reaction was in a 25 uL PCR mix using the Premix Ex Taq Kit (Vazyme, China). PCR conditions were as follows: denaturation at 94 °C for 2 min, then 30 cycles of 94 °C for 30 s, 60 °C for 30 sec, and 72 °C for 20 s; finally, an extension step for 7 min (ABI System). Each sample was run in triplicate for each marker.

### 2.3. qPCR

mRNAs’ expressions were measured by qPCR in this study, *eBD-1*, *ass BD-1*, *mule BD-1*, *TLR2*, *IL-1β*, *CCL2*, *CCL7*, *CXCL10*, *p65* relative to the *GAPDH* mRNA levels using the 2^−^^△△ct^ (i.e., ΔCT1 = CT treatment group − CT GAPDH, ΔCT2 = CT control group- CT GAPDH, and ΔΔCT = ΔCT1 − ΔCT2). Primers are shown in Table 2 [22,23], and synthesis in the Invitrogen Company (China). The qPCR conditions were as follows: denaturation at 95 °C for 10 min, then 40 cycles of 95 °C for 15 sec and 60 °C for 30 s (ABI Quantstudio™7), and qPCR operation following MIQE standard [24]. Each sample was run in triplicate for each marker.

### 2.4. Rapid Amplification of cDNA Ends (RACE) PCR 

A 5′ and 3′-RACE method was used to amplify the full-length sequence of *eBD-1*, *ass BD-1*, and *mule BD-1*, following the instructions of the SMARTer-RACE kit (Takara, China). Here, we designed new RACE primers for *eBD-1*, *ass BD-1*, and *mule BD-1* (Table 2). The primers P1, P2, and P3 were used to amplify the 5′ extremity of *eBD-1 ass BD-1* and *mule BD-1*, whereas the P4, P5, P6, and P7 primers were used to amplify the 3′ extremity. The amplified products were sent to Invitrogen (China) for sequencing.

### 2.5. The OEBD-1 and ELP-(V)_30_ Fragment Connection 

The Equus homolog full ORF was amplified using the restriction enzyme cutting site EcoRI, with the forward primer (5′-CCGGAATTC(EcoRI)ATGAGGGCACCCTACTT-3′) and the reverse primer (5′-CCCAAGCTTGAGGCAGCACTTGGCCTTCCC-3′). The ELP-(V)_30_ (GenBank No. KP756910.1) was amplified within the NotI restriction enzyme cutting site with the forward primer (5′-TGGGTACCGGGAATGGGTGT-3′) and the reverse primer (5′-TTTATAGCGG(NotI)TTACCCCATACCGGGAACACC-3′). The PCR conditions were as follows: denaturation at 94 °C for 5 min, then 30 cycles of 94 °C for 15 s and 66 °C (or 58 °C for ELP-V_30_) for 30 s (ABI System). The purified amplicon was cloned into the pMD19-T, with the forward primer (5′-CCGGAATTC(EcoRI)ATGAGGGCACCCTACTT-3′), reverse primer (5′-TTTATAGCGG(NotI)TTACCCCATACCGGGAACACC-3′), and a linker (5′-CGGCGGGAAGGCCAAGTGCTGCCCATTCCCGGTACCCACAT-3′). Overlap PCR conditions were as follows: denaturation at 94 °C for 5 min, then 30 cycles of 94 °C for 15 s and 60 °C. The purified amplicon cloned in the pMD19-T used EcoRI and NotI restriction enzymes to achieve the C-terminus ELP-(V)_30_-OEBD-1 N-terminus fragment.

### 2.6. The ELP-(V)_30_-OEBD-1 Recombination Proteins Expression In Vitro 

The ELP-(V)_30_-OEBD-1 recombinant protein was expressed in a pET-22b vector (Invitrogen, China) with the N-terminal 6-Gly His tag, according to the manufacturer’s instructions. In addition, the recombinant protein was expressed using the *E. coli* Top10 (Thermo Fisher, USA). The ELP-(V)_30_-OEBD-1 recombination protein was purified using the Inverse Transition Cycling (ITC), following the ELP-(V)_30_ protein purified assay described by Shamji et al. and Wu et al. [25,26], with slight modification. 

The ELP-(V)_30_-OEBD-1 protein supernatant was centrifugalized at 5500× *g* for 15 min after being placed in a water bath at 32 °C for 30 min. Next, the 2M NaCl (Sigma Aldrich, China) was added to the supernatant and centrifuged at 12,000× *g* at 15 °C for 15 min to harvest the sediment. Next, the collected precipitate was added with 10 mM Tris-HCl pH 8.5 (Sigma Aldrich, China) and placed at 4 °C for 30 min; then centrifuged at 12,000× *g* at 4 °C for 15 min. These steps were repeated twice. The expression of His-tagged protein was confirmed by western analysis using the horseradish peroxidase (HRP)-conjugated mouse-anti-His tag antibody (Invitrogen, China) (1:2000 dilution). The molecular weight of protein bands on the Western blot was estimated based on the broad range pre-stained protein marker (Transgen, China) on the imaging system (Bio-Rad, China).

### 2.7. Mice Mononuclear Macrophage J774A.1 Cell Line Culture

The J774A.1 macrophage cell line (purchased from China Center for Type Culture Collection) was cultured in DMEM with 10% FBS (Gibco, Shanghai, China), 1% penicillin, and 1% streptomycin (Gibco, China). The culture was incubated at 37 °C with 5% CO_2_. A new culture medium was added daily to replace old one. Furthermore, the cells were monitored until the number of J774A.1 was more than 80 percent in one microscopic field. Lastly, the cells were scraped and placed into 6-well with culture medium overnight, ready for succeeding experiments.

### 2.8. Preparation of the S. aureus-Infected J774A.1

We used the well-characterized *S. aureus* strain SA113 stored and maintained at the Inner Mongolia Agriculture University, as previously described [27,28]. The SA113 was cultured in a trypticase soya broth (TSB, Oxoid, UK) at 37 °C aerobic conditions with shaking at 50× *g* for 16 h. The *S. aureus* SA113 strain was collected after centrifugation at 5000× *g* for 5 min and ten-fold diluted using PBS. The SA113 count at 3 × 10^7^ CFU/mL in a trypticase soy agar (TSA, Oxoid, UK) was used for J774A.1 macrophage cell line infection.

J774A.1 cells were cultured overnight using the culture media DMEM with 10 % FBS, 1% penicillin, and 1% streptomycin. The cell culture was set up in a 6-well plate. Then, the culture media was replaced with DMEM with 10% FBS, and the SA113 was introduced for cell infection. The infected cells were either treated with PBS (uninfected or control groups), PET-22b empty vector or ELP-(V)_30_-OEBD-1.

### 2.9. Macrophage (J774A.1) Phagocytosis Assay

J774A.1 cells were placed in 6-well plates (1 × 10^6^ cells/well), washed 3 times with DPBS and *S. aureus* SA113 strain suspensions (MOI = 40:1) added to cell culture medium. Infected J774A.1 cells were incubated with lysozyme (at concentration 20 μM Sigma-Aldrich, China) at 37 °C for 1 h. After 0 h (uninfected or control group), 4 h and 6 h, the J774A.1 cells were washed 3 times with DPBS and lysed with a mixture of 0.25% trypsin and 0.025% Triton X-100 (Solarbio, China). Cell lysate (10 μL) was used for CFU assay and CFUs determined by TSA plate counting as follows: the lysate was diluted 100-fold in PBS and dropped onto TSA incubation at 37 °C for 16 h.

### 2.10. Enzyme-Linked Immunosorbent Assay (ELISA)

The culture was collected and centrifugated at 10,000× *g* for 5 min. The collected suspension was diluted five times in PBS. The detection of IL-6 (assay range: 9.6–223.4 pg/mL) and TNF-α (assay range: 416.5–995.4 pg/mL) was performed using ELISA kits (Mei Biao Biology, China). Each sample was run three times for each measurement.

### 2.11. Protein Extraction

J774A.1 protein extractions were obtained using Cell Protein Extraction Lysis Buffer (Thermo Fisher Scientific, Waltham, MA, USA) with Phosphatase Inhibitor (Thermo Fisher Scientific, USA). Cells were resuspended by 200 uL lysis buffer, and centrifugation at 14,000× *g* for 10 min at 4 °C then placed at room temperature 2 min. Protein was collected and stored in −80 °C used to Western blotting.

### 2.12. Coomassie Brilliant Blue Staining

Extracted proteins were standardized following Pierce BCA Protein Assay Kit (Thermo Fisher Scientific, USA) manufacturers’ instructions to 1 ug/μL using a spectrophotometer (Effendorf, Hamburg, Germany) and separated on a 12% SDS-PAGE gel with 20 μg/lane. Following electrophoresis, the SDS-PAGE was stained by Coomassie brilliant (Beyotime, China) for 15 min, and washed four times for 5 min in distilled water. 

### 2.13. Western Blotting

Protein sample concentrations were measured by BCA assay kit and 20 μg/mL subjected to Western blotting (WB), following electrophoresis in 12% SDS-PAGE. WB was conducted as follows: protein samples were blocked onto an immobilon-polyvinylidine difluoride (PVDF) membrane (Immobilon, Germany) by Bio-Rad Mini-Protran electroblotting system (Bio-Rad, China). Membrane was blocked with 5% skimmed milk powder (BD, USA) for 2 h at room temperature. Membranes were probed with antibodies for GAPDH, AKT, Syk, IκB-α (Proteintech, China) phosphorylated (P)-AKT P-Syk and P- IκB-α (CST, USA). AKT primary antibodies were used at 1:3000 dilution, GAPDH primary antibodies at 1:10,000 dilution and others at 1:1000 dilution, in primary antibody dilution reagent (Beyotime, Shanghai, China). Samples were incubated overnight at 4 °C and washed 3 times with TBST (Solarbio, Beijing, China) for 10 min. Membranes were incubated with secondary antibodies at 1:3000 dilution for 1 h, including HRP-linked goat anti-mouse IgG (CST, USA), HRP-linked rabbit anti-goat IgG (CST, USA), HRP-linked goat anti-rabbit IgG (CST, USA), then washed 4 × 5 min with TBST. To strengthen the chemiluminescence, membranes were visualized by ECL (Beyotime, China).

### 2.14. Immunofluorescence Assay

J774A.1 cells were placed in laser confocal culture dish (1 × 10^6^ cells/dish), stained with 8 µM 1,1-dioctadecyl-3, 3′-3′-3′ methylin- docarbocyanine perchlorate (DiI, Thermo, USA) and fixed with 4% paraformaldehyde at room temperature for 30 min. Stained J774A.1 cells were infected by SA113 or SA113 with Hoechst-labeled (MOI = 40:1) for 30 min in the dark, before incubation at 37 °C with 5% CO_2_. After 4 h, cells were washed 3 times with DPBS and a mixture of PBS and 0.025% Triton X-100 used for lysis. Cells were examined under laser scanning confocal microscope (ZEISS, Oberkochen, Germany) and fluorescence intensity analyzed. A single J774A.1 cell output in 3D image using the same grayscale value was produced.

### 2.15. Data Analysis 

Data were analyzed using the 1-way ANOVA followed by Tukey’s multiple comparisons test with Bonferroni’s post-test analysis of different treatments in J774A.1. Results were presented as mean ± SDs. Statistical significance was classified as follows: * *p* < 0.05; ** *p* < 0.01; *** *p* < 0.001, ns: *p* > 0.05. The sequence homology was analyzed using MEGA-6, All analyses were performed in GraphPad Prism 6.0 software (GraphPad company, 2012, San Diego, CA, USA).

## 3. Results

### 3.1. BD-1 Expression in Various Tissues of the Horse, Ass, Mule, and Homology Analysis

BD-1 is widely expressed in equines. To supplement BD-1 expression in other equid species, we used different tissues from the horse, ass, and mule to compare BD-1 gene expression levels and distribution. The results showed that BD-1 expression was present in the tongue, esophagus, trachea, lung, stomach, cardiac orifice, pyloric orifice, rectum, orchis, kidney, and skin (Figure 1a). Furthermore, the qPCR analysis showed that the *eBD-1* and *mule BD-1* are expressed in the trachea, stomach, heart, and skin, whereas the *ass BD-1* covered more organs, including the liver and kidney. Of note, the *mule BD-1* expression levels in the trachea, stomach, heart, and skin were significantly higher than in horse and ass (Figure 1b–d). To further examine the difference between *eBD-1* (NCBI submission No. #2613292), *ass BD-1* (NCBI submission No. #2614438), and *mule BD-1* (NCBI submission No. #2614440), we used RACE PCR to assess the entire sequence length of the genes. We observed that *ass BD-1* and *mule BD-1* are homologous to *eBD-1* at 98.9% and 97.5%, respectively (Figure 1e). 

### 3.2. The ELP-(V)_30_-OEBD-1 Recombination Protein Is Highly Expressed after Two Inverse Transition Cycling (ITC)

Here, we first confirmed the sequence of ELP-(V)_30_-OEBD-1 (for short OEBD-1) at 528bp. Similar to Shamji et al., we observed that the ELP-(V)_30_ is highly expressed at 32 °C [25]. To determine the effect of ITC on OEBD-1 expression, we measured the levels of OEBD-1 after different ITC repetitions through Coomassie brilliant blue staining and western-blotting (WB). The OEBD-1 recombination protein was predicted to have a molecular weight of 19,360 Da. Protein bands were identified SDS-PAGE gel in 15 kDa~25 kDa by Coomassie brilliant blue stained (Figure 2a), and the WB results showed that the OEBD-1 recombination protein expression is higher after two ITCs than a single ITC (Figure 2). However, when we added more ITC, the OEBD-1 expression levels were weaker after three ITCs and eventually disappeared after four ITCs. 

### 3.3. The OEBD-1 Recombination Protein Induces Various Cytokines Expression in SA113-Infected J774A.1 Cells

To determine whether the OEBD-1 is involved in the innate immune response in macrophage, we treated SA113-infected J774A.1 cells with 10 ug/mL OEBD-1 recombination protein and 10 ug/mL PET-22b empty vector recombination for 2 h, 4 h, and 6 h. The OEBD-1 recombination protein significantly increased the secretions of IL-6 and TNF-α after 6 h of treatment compared to the SA113-infected strain (*p* < 0.001, Figure 3a,b). Those treated with the PET-22b empty vector showed no significant difference in terms of IL-6 and TNF-α secretion in J774A.1 cells, in comparison with SA113-infected cells (*p* > 0.05, Figure 3a,b). Further, we measured other inflammatory associated factors mRNA expression. Similarly, the OEBD-1 had markedly increased the mRNA expression levels of *TLR2*, *IL-1β*, *CCL2*, *CCL7*, *CXCL10* and *NF-κB p65* mRNA expression after 6 h, in comparison with SA113-infected groups (*p* < 0.01, Figure 3c–h).

### 3.4. The OEBD-1 Recombination Strengthens Macrophage Phagocytosis of SA113

To further investigate the role of the OEBD-1 during *S. aureus*-infected macrophage, we used a TSA counting assay to show the rate of macrophage phagocytosis of *S. aureus*. We observed that the OEBD-1 recombination protein markedly strengthened the ability of J774A.1 to phagocytose the SA113 compared with the SA113-infected groups after 4 h (*p* < 0.001, Figure 4a). We did not observe a significant impact of PET-22b empty vector during treated SA113-infected J774A.1 cells (*p* > 0.05, Figure 4a). Furthermore, we evaluated the effect of OEBD-1 recombination protein treatment in SA113-infected J774A.1 through fluorescence assay and 3D imaging. We observed that the OEBD-1 significantly increased the fluorescent value of *S. aureus* in J774A.1 cells (*p* < 0.001, Figure 4b–f). The fluorescence results also illustrated that OEBD-1 enhanced macrophages phagocytosis *S. aureus* in vitro.

### 3.5. The OEBD-1 Recombination Promotes the Phosphorylation of Syk, AKT and IκB-α in SA113-Infected Macrophage

As aforementioned, the OEBD-1 enhanced cytokines expression during SA113 infection, and promoted phagocytosed *S. aureus* occurring in the macrophage J774A.1. To clarify the associated signaling pathway of the processes, we used Syk-PI-3K-ATK pathway, which involves phagocytosis and cytokines secretion for macrophage [18,19,20]. Additionally, the NF-κB pathway was closely associated with cytokines’ expression when the pathogenic microorganism infected the cell in vitro [30,31,32]. Thus, the phosphorylation of Syk, AKT and IκB-α were measured by Western blot (Original figures see Appendix A). We treated SA113-infected J774A.1 cells with 10 ug/mL OEBD-1 recombination protein for 15 min, 30 min, and 60 min. The results showed that SA113 activated Syk, ATK and IκB-α expression (Figure 5a–d), as expected, the OEBD-1 increased the phosphorylation of Syk, AKT and IκB-α after 15 min of treatment, compared to the SA113-infected cells (Figure 5a–d).

## 4. Discussion

Here, we show that the expression levels of Equus BD-1 are significantly abundant in the trachea and skin, as well as in tissues rich in the epithelial cells. Our finding is concordant to the conclusion that airway epithelial cells play an essential role in producing defensins [33]. Our data suggest that Equus BD-1 is expressed more in the heart tissue than other tissues, demonstrating that cardiomyocytes are a source of BD-1 [34]. The studies indicate that BD-1 is a local measure to protect an organ against a danger signal associated with infection [34,35]. With mules showing more organs producing BD-1 than horses and asses, our results may suggest that hybrids may have acquired a selection advantage against pathogenic microorganisms through β-defensins. 

Furthermore, we used RACE PCR to achieve the full-length gene sequences of *eBD-1*, *ass BD-1*, and *mule BD-1*, the data supplemented BD-1 sequences in other equids. The sequencing homology analysis of ORFs indicates that *eBD-1*, *ass BD-1*, and *mule BD-1* are identical, suggesting that Equus BD-1 ORF is conserved across the equine family. Thus, clarifying the role of the ORF in immune system is crucial for understanding the impact of effect of Equine BD-1 on innate immune responses. Based on this, we used *E. coli* expression system to express the ORF sequence, and explored its role in SA113-infected J774A.1 cells. 

BD-1 is often involved in response to TLR-mediated production of proinflammatory cytokines [15]. In our study, the OEBD-1 increased *TLR2* mRNA expression in SA113-infected J774A.1 cells. *TLR2* plays an important role in cytokine secretion during *S. aureus* infection [36,37]. In here, OEBD-1 enhanced cytokine mRNA expression in accordance with macrophage *TLR2* activation. Furthermore, the OEBD-1 activated phosphorylation of PI3K-AKT and IκB-a, and *p65* mRNA expression, suggested that OEBD-1 up-regulated pro-inflammation factors and chemokine, indicating a positive correlation with the PI3K-AKT-NF-κB pathway. Lin et al. showed that LPS induces cytokines expression on macrophage depending on the PI3K-NF-κB signaling pathway [30]. They also demonstrated that PI3K-NF-κB pathway is essential in PAMPs-infected macrophage.

Here, we showed that OEBD-1 can enhance the cytokines expression in SA113-infected J774A.1 cells. Macrophage phagocytosis is an enormously complex process that facilitates pathogen clearance and resolution of infections [20]. The Syk activation is an essential step for macrophage phagocytose PAMPs [19]. Activation of the Syk kinase results in transcriptional activation, cytoskeletal rearrangement, and the release of inflammatory mediators [38]. Moreover, the process depends on PI-3K, which transduces signaling. We showed the OEBD-1 promoted phosphorylation of Syk and AKT in J774A.1 cells during SA113 infection. Data from this report have shown that OEBD-1 promoted J774A.1 phagocytosis in SA113 via the Syk-PI3K-AKT pathway. In addition, chemokines involve macrophage recruitment to target tissue for phagocytosis [39,40]. We also showed that OEBD-1 increased chemokines *CCL2* and CXCL-10 mRNA expression in SA113-fected J774A.1 cells. We mapped the possible figure according to the results to illustrate the role of OEBD-1 in *S. aureus*-infected macrophages. Firstly, OEBD-1 enhanced *TLR2* expression, the signaling transduction to Sky-PI3K-AKT and promoted macrophage phagocytosis of *S. aureus*. Additionally, nuclear signal transduction eventually enhanced NF-kB (*p65*, IkB-a,) the expression of pro-inflammatory cytokines and chemokines upregulation (Figure 6). Our research provides a theoretical basis for the development of defensins as potential alternatives to antibiotics.

## 5. Conclusions

In summary, the BD-1 gene shows diverse tissue localization and expression levels in three equine species. Our results also support the idea that the family of peptides is evolutionarily conserved. In addition, the OEBD-1 recombination protein induces the secretion of pro-inflammatory cytokines and the ability of J774A.1 to phagocytose *S. aureus*. This study is the first report that shows the ORF of Equus BD-1, which promoted macrophage phagocytosis of *S. aureus*. The reported findings demonstrate that the recombinant OEBD-1 protein induced innate immune responses by triggering proinflammatory cytokine and chemokine expression while also enhancing macrophage phagocytosis of *S. aureus* mediated through the Syk-PI3K-AKT signal transduction pathway.

## Figures and Tables

**Figure 1 animals-12-02958-f001:**
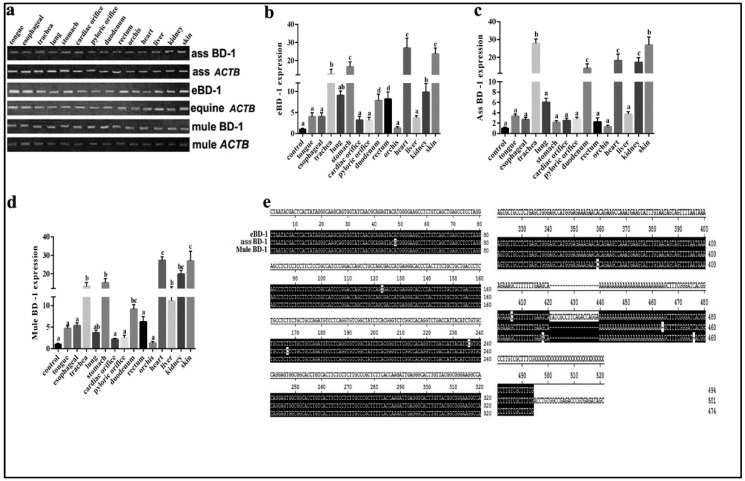
BD-1 expression in different tissues collected from horse, ass, and mule. (**a**) PCR-detected BD-1 expression in tongue, esophagus, trachea, lung, stomach, cardiac orifice, pyloric orifice, rectum, orchis, kidney, and skin. β-actin (ACTB) was used as the reference housekeeping gene. qPCR was used to measure different concentrations in various tissue, (**b**) *eBD-1*, (**c**) *ass BD-1*, (**d**) *mule BD-1*. (**e**) RACE PCR sequencing homology results. The analysis of sequence homology used MEGA-6 following the methods of Tamura K et al. [29]. Significant differences among results were determined using one-way ANOVA, followed by Dunnett’s test to control the comparison number (*n* = 3). Data are presented as means ± SD. Different letters indicate significantly different means at *p* < 0.05.

**Figure 2 animals-12-02958-f002:**
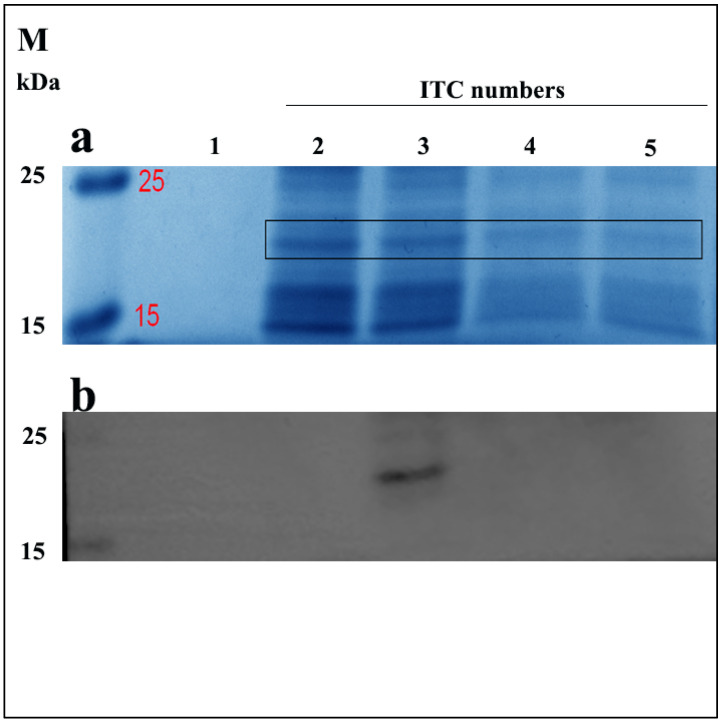
Expression of recombinant OEBD-1 in the *E. coli* Top10. Analysis of crude expression product in SDS-PAGE gel by Coomassie brilliant blue staining (**a**), and WB (**b**) with anti-6 × Gly-His. Purified OEBD-1 (∼19 kDa), the expressed OEBD-1 is indicated in gel by the filled black pane 1: pET-22b vector, 2: target protein was purified after once ITC, 3: target protein was purified after twice ITC, 4: target protein was purified after three times ITC, 5: target protein was purified after four times ITC, M: protein marker.

**Figure 3 animals-12-02958-f003:**
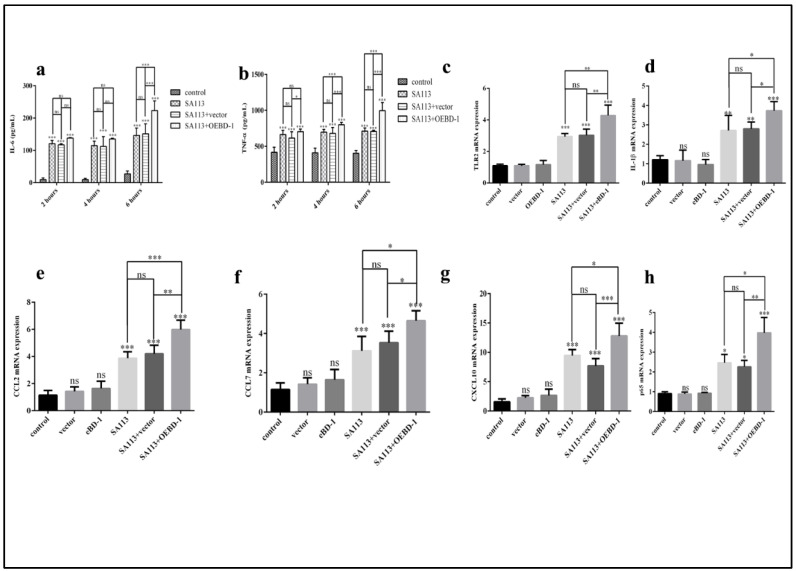
J774A.1 cells were infected with *S. aureus* SA113 strain (MOI = 40:1) for 2 h, 4 h, and 6 h. IL-6 and TNF-α secretion were measured by ELISA (**a**,**b**). The results showed that IL-6 and TNF-α expression were both highest in the SA113-infected strain after 6 h. Therefore, we investigated *TLR2* (**c**), *IL-1β* (**d**), *CCL2* (**e**), *CCL7* (**f**), *CXCL10* (**g**), *NF-κB p65* (**h**) mRNA expression after 6 h by qPCR. * *p* < 0.05, ** *p* < 0.01, *** *p* < 0.001, ns: no significant (*p* > 0.05).

**Figure 4 animals-12-02958-f004:**
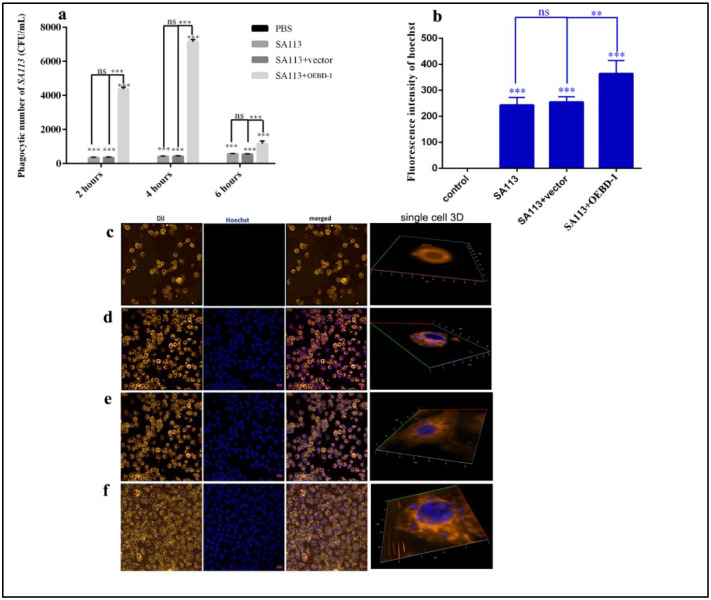
The numbers of J774A.1 phagocytoses SA113 counted by TSA culture (**a**). The fluorescent values of SA113 were displayed in the ZEISS system (**b**), and the data were achieved in three random visions (*n* = 3). The single-cell 3D imaging was showed J774A.1 phagocytose SA113 in confocal microscopy, J774A.1 cells were marked by orange fluorescent, and SA113 was marked by blue fluorescence (**c**–**f**); (**c**) control or uninfected group, (**d**) SA113-infected group, (**e**) SA113+vector group, (**f**) SA113+*eBD-1* recombination protein. ** *p* < 0.01, *** *p* < 0.001, ns: no significant (*p* > 0.05).

**Figure 5 animals-12-02958-f005:**
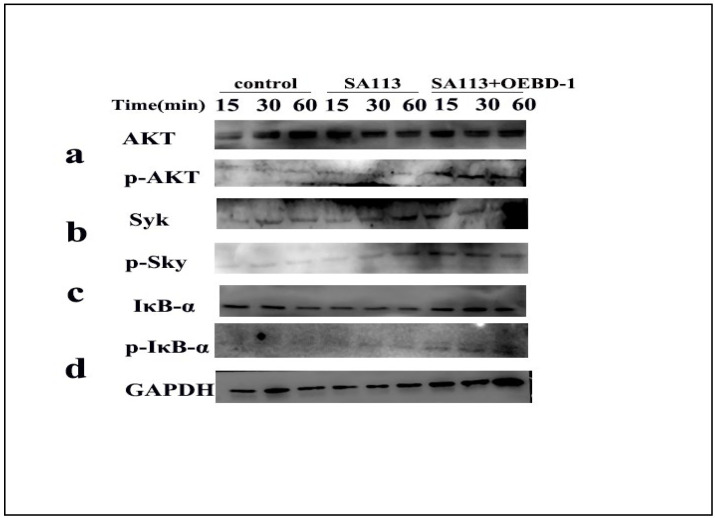
Macrophage J774A.1 in vitro were infected with SA113 at MOI = 40:1 or SA113 with 10 ug/mL OEBD-1 for 15 min, 30 min, and 60 min. Western blotting was used to assess GAPDH (**d**), non-phosphorylated and phosphorylated AKT (**a**), Syk (**b**), and IκB-α (**c**).

**Figure 6 animals-12-02958-f006:**
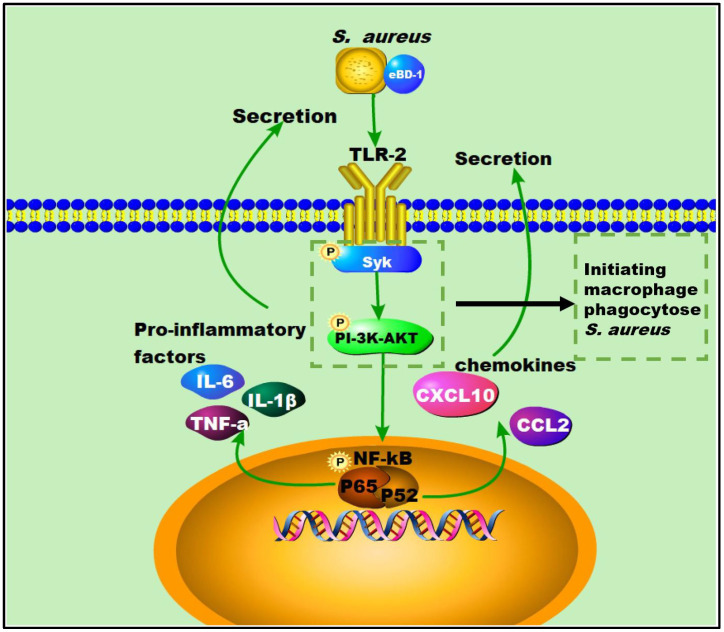
The possible figure according to the results, to illustrate the role of OEBD-1 in *S. aureus*-infected macrophage, the OEBD-1 promotion macrophage phagocytosed *S. aureus* via Syk-PI-3K-AKT pathway. Furthermore, OEBD-1 activated *TLR2* to upregulated cytokines and chemokines expression via Syk-PI-3K-AKT-NF-κB pathway.

**Table 1 animals-12-02958-t001:** The primer messages of PCR and qPCR.

Primer Name	Forward (5′-3′)	Reverse (5′-3′)	GenBank Accession No.
equine β-actin	GGCTCCCAGCAGATGAA	GCATTTGCGGTGGACGAT	AF035774.1
*eBD-1*	CAGGTGTCGGCTATCTCACG	CCTTCCCGCCGTAACAAGT	XM_005606422.3
*ass BD-1*	ATGTCCTCAGGTGTCGGCTA	ACAAGTGCCCTCAATCTTGGT	XM_014857440.1
ass β-actin	TGCGTGACATCAAGGAGAAG	ACAGGTCCTTACGGATGTCG	XM_014853634.1
*mule BD-1*	CAGGTGTCGGCTATCTCACG	CCTTCCCGCCGTAACAAGT	KP710585
mule β-actin	CCAAGGGTGGATCCTTA	AGGAGGAATGGGAATATT	KU947963
*TLR2*	TTTGCTCCTGCGAACTCC	GCCACGCCCACATCATTC	XM_006501460.4
*IL-1β*	ACCTTCCAGGATGAGGACATGA	CTAATGGGAACGTCACACACCA	AL808143.5
*CCL2*	ATCCACGGCATACTATCAACATC	TCGTAGTCATACGGTGTGGTG	XM_036154586.1
*CCL7*	CCACATGCTGCTATGTCAAGA	ACACCGACTACTGGTGATCCT	NM_013654.3
*CXCL10*	CAGTGAGAATGAGGGCCATAGG	CGGATTCAGACATCTCTGCTCAT	XM_021161764.2
*NF-κB P65*	TTCCCTCAGAGCCAGCCCAGG	AGCGGAATCGCATGCCCC	M61909.1
GAPDH	CAATGTGTCCGTCGTGGATCT	GTCCTCAGTGTAGCCCAAGATG	XM_036165840.1

**Table 2 animals-12-02958-t002:** The primer messages of RACE PCR.

Primer Name	Forward (5′-3′)
P1	CTAATACGACTCACTATAGGGCAAGCAGTGGTATCACGCAGAGT
P2	CTAATACGACTCACTATAGGGC
P3	CCGCCGTAACAAGTGCCCTCAATC
P4	(dT) n-CGAAAGCGACAAGGCCGTGATCCCGAAAGC
P5	CGAAAGCGACAAGGCCGTGATCCCGAAAGC
P6	GGTGTCGGCTATCTCACGGGTCTCG
P7	GCTATCTCACGGGTCTCGGCCACAGGT

## Data Availability

The sequences used in the research were uploaded to NCBI, the *eBD-1* (accession number OP265307), the assBD-1 (accession number OP265308), and the *mule BD-1* (accession number OP265309). In addition, the other data generated or analyzed during this study are included in this published article.

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
