# Peer review of "Equus β-Defensin-1 Regulates Innate IMMUNE Response in S. aureus-Infected Mouse Monocyte Macrophage"

_animals, 2022, doi:10.3390/ani12212958_

Round 1
Reviewer 1 Report
Summary:
The study focuses on defensin antimicrobial peptides from three different equus species from Asia. These antimicrobial peptides are commonly found in nasal airways secretions and some other parts of the body. The data obtained shows relevance to the previous studies.
About the article:
The importance of selected equus species has not been adressed well enough. What is the benefit for human? Please mention the importance of performing this study on these animals. The motivation behind the study should also be included in conclusion and discussion.
The article
Suggestions:
1. If it is not required by the journal guideline, please remove simple summary section.
2. Please give species names full first then use the abbreviations, follow this rule along with rest of the article.
3. When you say the sequences are found identical (line 15), please give some units or numbers, such as 99% or 90%.
4. In abstract you can add the name of the expression system used in E. coli.
5. Please rephrase the sentence in line 19-20. It is not correct grammatically.
6. Line 28-32, please rephrase or state in another format since it is not easy to follow and understand.
7. Line 40, use cationic instead of cation.
8. The secretion tissues of defensin take too much space in the article but the title does not mention about it. If it is not critical you can remove some of these info.
9. Table 2. should come after line 130.
10. In line 372, did you mean base"d" on?
11. Species names should be ithalic, line 412.
12. Line 413, it says "remarkable ORF" this is not a fact. Please try to use a proper scientific language.
Thanks
Reviewer 2 Report
This report provides evidence to demonstrate equine beta defensin-1 expression in equid species that have not been investigated previously. The authors demonstrate eBD-1 sequence conservation among equine hosts and demonstrate antimicrobial peptide function using an in vitro system. Investigative objectives are appropriate and well tested. There is a need for extensive editing to improve the clarity and accuracy of writing.
Specific comments:
Line 11, Beta-defensin 1is rich in disulfide bonds and is among antibacterial peptides with direct bactericidal activity.
Line 12-13, Equine BD-1 was initially reported in 2004, however, there are no reports on the specific mechanisms that characterize the impact of EBD-1 on innate immune function.
Line 16-17, Next we expressed the ORF of the Equus BD-1 sequence fragment in an E. coli expression system, and used this to treat an S. aureus infected murine macrophage cell line in vitro.
Line 19-20, This report provides a theoretical basis for the development of defensins as potential alternatives to antibiotics.
Line 21, Beta-defensin-1 (BD-1) is among the class of antimicrobial peptides that are rich in disulfide bonds have direct antimicrobial activity and show enhanced expression following external stimulation.
Line 28, The ORF encoding OEBD-1 was expressed in an E. coli expression system. This expression system was combined with S. aureus infected J774A.1 macrophage cell line to determine the influence on innate immune cell mediator expression.
Line 32, Using this expression model system, it was determined that OEBD-1 protein enhanced IL-6 and TNF-alpha secretion
Line 35, Equus BD-1 was identified to play an essential role in macrophage associated innate immune responses in an in vitro system.
Line 43, Previous work demonstrated that human BD-1(HBD-1) is abundantly
Line 61, Equine BD-1 (eBD-1) is widely expressed in a range of equine tissues. Recent reports have shown that eBD-1 is mildly chemotactic for equine neutrophils and monocytes
Line 63, whether eBD-1 affects macrophage function has not been reported.
Line 66, Cytokine expression and phagocytosis involves various signal transduction pathways within macrophages.
Line 68, elements for macrophage phagocytosis to occur.
Line 72, OEBD-1 in an E. coli expression
Line 77, OEBD-1 enhanced innate
Line 98, Cellular RNA extraction was performed following manufacturer instructions.
Line 110, was run in triplicate for each marker.
Line 118, Primers are shown
Line 260, in other equid species
Line 290, Protein bands were identified on SDS-PAGE gel
Line 291, and the WB results showed
Line 331, the authors should expand the statement that no only was there an increase fluorescent value of S. aureus in J774A.1 cells, but that this finding represents enhanced phagocytosis.
Line 372, the authors are reporting on the fact that the impact of enhanced EBD-1 expression. It should be stated that it is crucial (important) to understand the impact of effect of EBD-1 in innate immunity. Affection is an inappropriate word.
Line 378, OEBD-1 enhanced cytokine mRNA expression in accordance with macrophage TLR2 activation.
Line 386, Macrophage phagocytosis is an enormously complex process that facilitates pathogen clearance and resolution of infections.
Line 388, Activation of Syk kinase results in
Line 392, Data from this report has shown that OEBD-1 promoted J774A.1
Line 397, OEBD-1 enhanced TLR2 expression, signaling transduction to Sky-PI3K-AKT and promoted macrophage phagocytosis
Line 399, Additionally, nuclear signal transduction enhanced NF-kB (p65, IkB-a) expression of pro-inflammatory cytokine and chemokine upregulation.
Line 414, Reported findings demonstrate that recombinant OEBD-1 protein induced innate immune responses by triggering proinflammatory cytokine and chemokine expression while also enhancing macrophage phagocytosis of S. aureus mediated through the Syk-PI3K-AKT signal transduction pathway.
Round 2
Reviewer 2 Report
Overall, the revisions to the manuscript entitled Equus β-defensin-1 regulates innate immune response in S. aureus-infected mouse monocyte macrophages or cells are appropriate. Further editing will enhance quality of this writing.
Author Response
We wish to re-submit an original research article for publication in the journal of Animals. Thank you for your letter and comments concerning our manuscript entitled ‘Equus β-defensin-1 regulates innate immune response in S. aureus-infected mouse monocyte macrophage’ (Manuscript No. animals-1943168(revision)). Those comments are all valuable and very helpful for revising and improving our paper, as well as the important guiding significance to our research. We have modified the manuscript according reviewers' comments. We corrected the errors according to the reviewers as following:
Because of we used ‘Track Changes’ function, so the number of lines were including section of deletion and addition.
1. We have modified the statement in our paper to reduce the duplication rate. (Line 105-107, 125-126, 129-130, 212-219, 230-232, 262-276, 287-295).
2. Because of the statements are all in part of 'materials and methods', thus the references have no change.